# Exploring the impact of opioid use on outcomes in allogeneic hematopoietic stem cell transplantation

**Tommy Alfaro Moya**[1], **Abel Santos Carreira**[2], **Shiyi Chen**[3], **Mats Remberger**[4], **Refik Saskin**[5], **Igor Novitzky-Basso**[1]*, **Jonas Mattsson**[1]*

1 Hans Messner Allogeneic Transplant Program, Princess Margaret Cancer Centre, Toronto, Canada, 2 Hematology Department, Institut Catala D'Oncologia, Badalona, Spain, 3 Department of Biostatistics, Princess Margaret Cancer Centre, University Health Network, Toronto, Ontario, Canada, 4 Department of Medical Sciences, Uppsala University and KFUE, Uppsala University Hospital, Uppsala, Sweden, 5 Institute for Clinical Evaluative Sciences, ICES Toronto, Ontario, Canada

* igor.novitzkybasso@uhn.ca (IN-B); jonas.mattsson@uhn.ca (JM)

## Abstract

### Introduction

Hematological malignancies and allogeneic hematopoietic cell transplantation (alloHCT) often necessitate the use of opioids due to significant pain. This study aimed to investigate the impact of opioid use on the clinical outcomes of patients undergoing alloHCT.

### Methods

A retrospective cohort study was conducted by merging data from our local transplant database with anonymized pharmacy records obtained from the Institute for Clinical Evaluative Sciences (ICES). We analyzed 681 patients who underwent alloHCT at Princess Margaret Cancer Centre between January 2010 and December 2019. Patients who initiated opioid use within one-year post-alloHCT and had opioid prescriptions for more than 30 days were categorized as intense opioid users (IOU). Additionally, patients who started opioids before or within one-year post-alloHCT and had opioid prescriptions for less than 30 days but died while on opioids were also classified as IOU. The analytical code used for the analysis is available in the Supporting Information file.

### Results

Among the 681 patients, 51 were identified as IOU. The two-year overall survival (OS) was significantly lower in the IOU group, with 29.4% survival compared to 53% in non-IOU (HR 1.77, 95% CI 1.26–2.48, p = 0.0008). Multivariate analysis indicated that IOU status was associated with a 2.32 times higher instantaneous rate of death compared to non-IOU (HR 2.32, 95% CI 1.5–3.5, p = 0.002). The median time for relapse was 147 days in the IOU group (range 52–393) and 209 day for non-IOU (range 96–1793), p = 0.0082. Furthermore, the relapse rate at two years was notably higher in the IOU group (31.4% vs. 16.4%, p = 0.0049). The analysis of factors independently associated with relapse-free survival

**Data availability statement:** The dataset from this study is held securely in coded form at ICES. While legal data-sharing agreements between ICES and data providers (e.g., health-care organizations and government) prohibit ICES from making the dataset publicly available, access may be granted to those who meet pre-specified criteria for confidential access, available at www.ices.on.ca/DAS (email: das@ices.on.ca). The full dataset creation plan and underlying analytic code are available from the authors upon request, understanding that the computer programs may rely upon coding templates or macros that are unique to ICES and are therefore either inaccessible or may require modification to be used in other environments. Specific inquiries can be directed to the DAS team via email at das@ices.on.ca. Interested researchers can replicate our study findings in their entirety by obtaining access to the minimal dataset from ICES and following the methods outlined in our manuscript. The ICES DAS program provides a structured application process for researchers seeking access to ICES data, ensuring that all necessary conditions for data use are met. Additionally, the dataset creation plan and analytic code, available upon request, provide further guidance to enable replication. We confirm that the authors did not have any special access privileges to the data. All researchers who meet ICES's criteria for access and follow the application procedures can obtain the minimal dataset under the same conditions. Unfortunately, uploading the complete minimal anonymized dataset to a stable, public repository is not possible due to the legal data-sharing agreements governing ICES data.

**Funding:** The author(s) received no specific funding for this work.

**Competing interests:** The authors have declared that no competing interests exist.

(LFS) showed that IOU status, age, donor type, and cytogenetic risk were significant predictors. At two years, relapse-free survival was 29.4% in the IOU group compared to 52.5% in the non-IOU group (p < 0.001).

## Conclusions

In our study, we found a correlation between intense opioid use in alloHCT patients and worse overall survival, particularly concerning higher relapse rates. These findings highlight the need for further research into pain management strategies to improve outcomes and reduce potential toxicity.

## Introduction

Pain is a common and distressing symptom for patients with malignancies, characterized by an unpleasant sensory and emotional experience related to actual or potential tissue damage [1]. It affects approximately 55% of cancer patients and up to 65% of those with advanced disease [2].

Due to the absence of specific guidelines for hematological patients, pain management strategies are often adapted from those for solid tumor guidelines. Opioids are a primary treatment for cancer-associated pain [3]. The World Health Organization (WHO) recommends initiating opioids with or without non-opioid medications for moderate to severe pain [4]. While opioids play a crucial role in managing both acute and chronic pain, they carry significant risks. The WHO recommends a three-step analgesic principle, transitioning from non-opioid agents for mild pain, adding weak opioids for moderate pain, and finally incorporating strong opioids for severe pain, alongside non-opioid and adjuvant therapies at each stage [4,5].

Opioids are the most frequently prescribed analgesics in both the United States and Canada [6,7].

However, their overuse has contributed to a significant public health crisis, with opioid overdose deaths becoming a major cause of mortality ultimately reducing life expectancy in the United States [8]. According to the CDC, opioid doses greater than 50 morphine milligram equivalents (MME) per day significantly increase the risk of fatal overdose without offering pain relief benefits [9].

Allogeneic hematopoietic cell transplantation (alloHCT) is a potentially curative treatment for patients with various hematologic conditions, but it is frequently accompanied by painful syndromes that begin early in the transplant process [10,11]. Oral mucositis, the epithelial damage of the oral mucosa due to chemotherapy and radiation, is the most common source of pain in patients undergoing alloHCT [10,12,13]. While opioids are often utilized to manage this pain, the absence of standardized guidelines results in significant variability in their use [14,15]. We aimed to explore the potential impact of opioid use on clinical outcomes, including survival, incidence of relapse, and transplant-related mortality, in patients undergoing alloHCT for various conditions.

## Methods

The information from our clinical database was merged with a dataset obtained from ICES containing prescription details for opioids (pharmacy fill records including dosing, number of dispensed days, and date of dispensation) as of September, 2022. Pharmacy fill records served as a surrogate marker for the opioid use intensity and duration. All the direct identifiers were removed before data analysis. The study was approved by the University Health Network Research Ethics Board and adhered to the standards outline in the Declaration of Helsinki.

In this analysis, we included 1,207 adult patients who underwent allogeneic hematopoietic cell transplantation (alloHCT) between January 7, 2010, and December 31, 2019. The patients in the study had started opioid medication before alloHCT or within one year from transplant. Patients were classified as 'intense opioid users' (IOU) if they used opioids for more than 30 consecutive days within the first year following alloHCT. Patients who initiated opioid use before alloHCT or within one-year post-alloHCT and used opioids for fewer than 30 days were also classified as IOU if they died while still on opioids within this period.

To mitigate survival bias—arising from the delayed initiation of opioids in some patients—we standardized the start time across all groups. Survival bias occurs because IOU patients must have survived until opioid initiation, making comparisons unfair if the alloHCT date is used as the start time for the comparator group.

For opioid users, the start time was defined as follows:

For patients who began opioid use within one-year post-alloHCT with a total duration exceeding 30 days, the start time was the first instance of opioid use.

For patients who began opioid use before alloHCT or used opioids for fewer than 30 days post-alloHCT but died while on opioids within one year, the start time was the date of alloHCT. The median start time for opioid users was 95 days post-alloHCT.

For the comparator group, the start time was aligned with the opioid users' median start time to account for survival bias. Specifically:

For patients who never used opioids, the start time was defined as alloHCT + 95 days.

For patients who started opioids but did not meet the IOU criteria (e.g., used opioids for fewer than 30 days within one-year post-alloHCT and did not die while on opioids), the start time was defined as the time of opioid initiation + 95 days.

For patients who initiated and stopped opioid use before alloHCT, the start time was also set as alloHCT + 95 days.

This alignment ensured that equivalent time points relative to risk exposure were used for both groups. All predictors included in the Cox proportional hazards model, Kaplan-Meier analysis, and Gray's test were known at the defined start time (time 0).

The following exclusion criteria were applied (Fig 1):

Patients who underwent a second alloHCT (n=83).

Patients with non-linkable ID to ICES databases or incorrect data (n = 8). Patients with non-malignant diseases, including myeloma, DCN, GATA2 deficiency, MF, MPN, SAA, and other non-malignant conditions (n = 132).

After these exclusions, 984 patients remained. From this group:

Patients who initiated opioid use more than one year after alloHCT (n = 203) were excluded to avoid bias, as the highest risk for relapse occurs within the first year post-alloHCT.

Non-opioid users who died or had their last follow-up before 95 days post-alloHCT (n = 100) were excluded to align with the defined start time for opioid users.

This left a final sample size of 681 patients for survival analysis. For analyses of chronic GVHD (cGVHD) or graft-versus-host-free survival (GRFS), we further excluded patients who died or relapsed within the first 100 days post-alloHCT, resulting in a sample size of 514 patients.

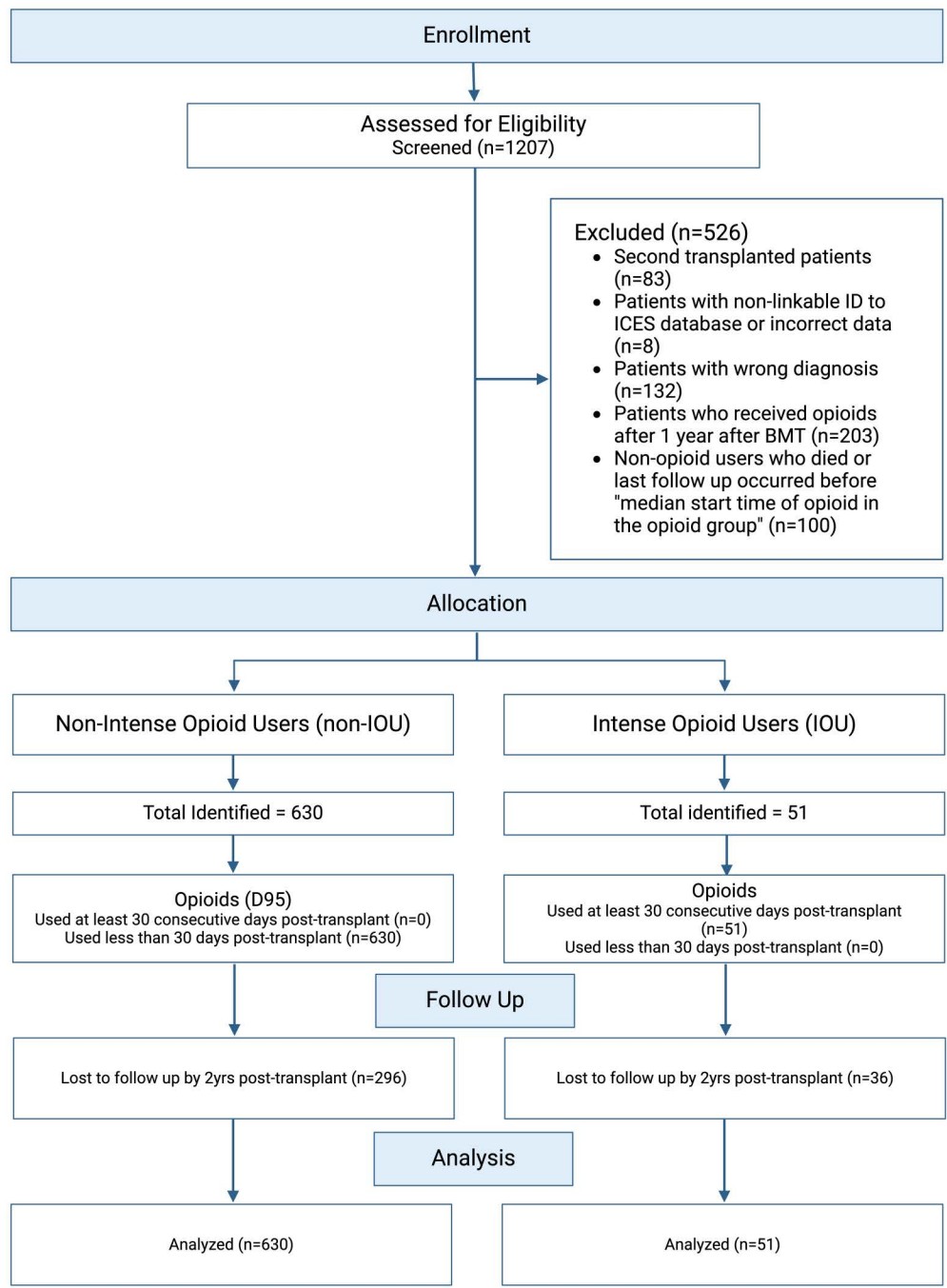

**Fig 1. Selection criteria for sample.**

## Conditioning regimen and graft versus host disease prophylaxis

Transplant conditioning regimens were categorized as myeloablative (MAC) or reduced-intensity conditioning (RIC) according to established criteria [16]. Among the patients, 25 (49.02%) in the IOU group and 267 (42.38%) in the non-IOU group received MAC, while 26 (50.98%) in the IOU group and 363 (57.62%) in the non-IOU group underwent RIC alloHCT.

The MAC regimen included fludarabine ($50\,mg/m^2$/day intravenously on days $-5$ to $-2$) and busulfan ($3.2\,mg/kg$/day IV on days $-5$ to $-2$). The RIC regimen consisted of fludarabine ($30\,mg/m^2$/day IV on days $-5$ to $-2$), busulfan ($3.2\,mg/kg$/day IV on days $-3$ and $-2$), and 200 cGy of total body irradiation on day $-1$.

GVHD prophylaxis regimens included rabbit ATG (Thymoglobulin, Genzyme-Sanofi, Lyon, France) in two dosage regimens. For unrelated donor transplants, a total dose of $2\,mg/kg$ was administered over days $-3$ and $-2$. For haploidentical donor transplants, a total dose of $4.5\,mg/kg$ was given across days $-3$, $-2$, and $-1$. In both regimens, post transplant cyclophosphamide (PTCy) was given at $50\,mg/kg$ on days $+3$ and $+4$, and cyclosporine A (CsA) was started at $2.5\,mg/kg$ every 12 hours beginning on day $+5$. For matched sibling donors (MSD), the regimen included PTCy ($50\,mg/kg$ on days $+3$ and $+4$), mycophenolate mofetil (MMF) at $15\,mg/kg$ every 8 hours from day $+5$ for 30 days, and CsA at $2.5\,mg/kg$ every 12 hours starting from day $+5$. An alternative regimen for patients ineligible for PTCy combined ATG with methotrexate (MTX) at $15\,mg/m^2$ on day $+1$, followed by $10\,mg/m^2$ on days $+3$ and $+6$. In this alternative regimen, CsA was introduced at $2.5\,mg/kg$ every 12 hours starting on day $-1$.

All patients received prophylactic antimicrobials, with cytomegalovirus and Epstein-Barr virus monitoring conducted weekly by polymerase chain reaction assay (PCR).

Cytogenetic risk was classified according to Grimwade et al [17]. We established two risk groups: low risk (assigned 0 or 1) for patients with low or intermediate cytogenetic risk, and high risk (assigned 2) for those with adverse cytogenetics.

## Statistical analysis

Descriptive statistics were generated for patient characteristics and clinical measurements. Continuous variables were summarized as means (with standard deviations or confidence intervals) or medians (range), depending on the data distribution. Differences between continuous variables were assessed using two-sample t-tests or Wilcoxon tests whereas categorical variables were compared using Chi-squared tests or Fisher's exact tests. Kaplan-Meier curves were plotted for outcomes including overall survival (OS), relapse free survival (RFS) and graft-versus-host disease (GVHD)-free relapse-free survival (GRFS). Differences in Kaplan Meier curves between opioid users and non-users were assessed using log-rank tests. Cumulative incidence curves were created for outcomes including relapse, non-relapse related mortality, aGVHD and cGVHD, with Gray's test used to evaluate differences between intensive opioid users and non-users.

Univariate Cox proportional hazard models were employed to examine the associations between OS, RFS, and GRFS, along with risk factors such as opioid user status (yes vs no), diagnosis, conditioning regimen and so on.

For multivariate analysis, the following clinical variable were included: conditioning regimen (RIC vs. MAC), age ($\geq 60$ yrs vs. $< 60$ yrs), disease risk index (DRI) (high/very high risk vs. intermediate vs. low risk), transplant period (2004–2008 and 2009–2013), GVHD prophylaxis (with or without T-cell depletion [TCD]), and HLA mismatched vs. matched donor. Hazard ratios (HRs) and 95% confidence intervals (CIs) were calculated for the clinical variables, and a stepwise selection process was performed for model selection using $p < 0.05$ as criteria for variable entry. A p-value less than 0.05 was considered statistically significant. All statistical analyses were performed using SAS 9.4 software (Cary, NC, USA).

## Results

### Patient characteristics

Study population characteristics are presented in Table 1. Most patients in both groups were in the age range of 40–59 years, with 22 patients (43.14%) in the IOU group and 283 patients

**Table 1. Characteristics of patients that underwent alloHCT in PMCC between the years 2010–2019.**

|  | Opioid user | Non-opioid user | P value |
|---|---|---|---|
| **N** | **51** | **630** |  |
| **Days to discontinuation of immunosuppressant** |  |  |  |
| Median (range) | 102.5 (21 to 1393) | 109.5 (−253 to 2775) | 0.71 |
| **Age** |  |  | 0.20 |
| 18–39 | 15 (29.41%) | 123 (19.52%) |  |
| 40–59 | 22 (43.14%) | 283 (44.92%) |  |
| 60 + | 14 (27.45%) | 224 (35.56%) |  |
| **Diagnosis** |  |  |  |
| ALL/MPAL | 8 (15.69%) | 77 (12.22%) | 0.15 |
| AML | 23 (45.10%) | 355 (56.35%) |  |
| CMML/MDS | 7 (13.73%) | 105 (16.67%) |  |
| Others(CML/CLL/Lymphoma) | 13 (25.49%) | 93 (14.76%) |  |
| **Donor** |  |  | 0.24 |
| MM (Haplo or MM URD) | 15 (29.41%) | 136 (21.59%) |  |
| MRD | 19 (37.25%) | 213 (33.81%) |  |
| MUD | 17 (33.33%) | 281 (44.60%) |  |
| **Gender** |  |  | 0.33 |
| Female | 20 (39.22%) | 292 (46.35%) |  |
| Male | 31 (60.78%) | 338 (53.65%) |  |
| **Stage** |  |  | 0.19 |
| CR1 | 21 (65.63%) | 347 (75.6%) |  |
| CR2 + CR3 + partial response | 11 (34.38%) | 112 (24.4%) |  |
| **RIC/MAC** |  |  | 0.36 |
| MAC | 25 (49.02%) | 267 (42.38%) |  |
| RIC | 26 (50.98%) | 363 (57.62%) |  |
| **Molecular** |  |  | 0.88 |
| neg | 34 (73.91%) | 392 (72.86%) |  |
| muted | 12 (26.09%) | 146 (27.14%) |  |
| **Cytogenetics** |  |  | 0.06 |
| 0–1 | 34 (66.67%) | 400 (80.65%) |  |
| 2 | 17 (33.33%) | 96 (19.35%) |  |
| **KPS** |  |  | 0.30 |
| 90–100 | 26 (89.66%) | 348 (82.08%) |  |
| **HCT-CI** |  |  | 0.40 |
| 0 | 10 (30.30%) | 101 (22.10%) |  |
| 1–2 | 10 (30.30%) | 187 (40.92%) |  |
| 3 or above | 13 (39.39%) | 169 (36.98%) |  |
| **DRI** |  |  | 0.96 |
| 0–1 | 25 (75.76%) | 354 (77.46%) |  |
| 2–3 | 8 (24.24%) | 103 (22.54%) |  |

AML, acute myeloid leukemia; ALL, acute lymophoblastic leukemia; MPAL, mixed phenotype acute leukemia; CMML, chronic myelomonocytic leukemia; Cytogenetics: risk as established by Grimwade et al., 0 favorable, 1 intermediate, 2 adverse [17,18]; MDS, myelodysplastic syndromes; CML, chronic myeloid leukemia; CLL, chronic lymphocytic leutkemia; RIC, reduced-intensity conditioning; MAC, myeloablative conditioning; MM, multiple myeloma; MRD, minimal residual disease; Molecular: Mutation status defined by the presence/absence of FLT3 ITD/TKD or NPM1 mutations by PCR; URD, unrelated donor; KPS, Karnofsky performance status; HCT-CI, hematopoietic cell transplantation comorbidity index; DRI, Disease Risk Index; CR, complete remission.

(44.92%) in the non-IOU group. Females comprised 20 patients (39.2%) in the IOU group and 292 patients (46.3%) in the non-IOU group.

Acute myeloid leukemia (AML) was the most common indication for alloHCT in both groups, affecting 23 patients (45.1%) in the IOU group and 355 patients (56.3%) in the non-IOU group. Among the IOU group patients, 17 (33.33%) were in the high-risk cytogenetic group compared to 96 (19.35%) in the non-IOU group (p = 0.06). Most patients in both groups were transplanted after achieving first complete remission (CR1), with 21 patients (65.6%) in the IOU group and 347 patients (75.6%) in the non-IOU group.

Opioids were initiated based on provider prescriptions in response to clinical indications such as moderate to severe pain related to post-transplant complications, including mucositis, musculoskeletal pain, or GVHD. The median time to opioid initiation in the IOU group was 95 days post transplant, ranging from 305 days before transplant to 304 days after transplant, and 13.7% of these patients started opioids within the first 30 days post-alloHCT. Notably, 32 out of 51 IOU patients had overlapping opioid prescriptions, indicating multiple active prescriptions simultaneously. The median time to discontinuing immunosuppression was day 102 for the IOU group and day 109 for the non-IOU group (p = 0.71).

A summary of the multivariate analysis for the main post alloHCT outcomes is presented in Tables 2 and 3.

**Table 2. Univariate analysis of selected factors:** NRM, nonrelapse mortality; RFS, relapse-free survival; GRFS, GvHD-free relapse-free survival; UVA, univariate analysis; KPS, Karnofsky performance scale; MAC, myeloablative conditioning; RIC, reduced intensity conditioning; DRI, disease risk index; HCT-CI, hematopoietic cell transplant comorbidity index; CR1, complete remission 1; HR, hazard ratio; BM, bone marrow; PBSCm peripheral blood stem cells.

| Predictor | Mortality | | NRM | | RFS | | GRFS | |
|---|---|---|---|---|---|---|---|---|
| | HR (95% CI) | p value | HR (95% CI) | p value | HR (95% CI) | p value | HR (95% CI) | p value |
| Opioid User (Yes vs No) | 1.77 (1.26–2.48) | <0.001 | 1.20 (0.77–1.87) | 0.41 | 1.73 (1.24–2.42) | 0.001 | 1.59 (1.06–2.37) | 0.024 |
| Group age at BMT (18–39 vs ≥60) | 0.47 (0.34–0.64) | <0.001 | 0.43 (0.28–0.65) | <0.001 | 0.49 (0.36–0.67) | <0.001 | 0.83 (0.61–1.13) | 0.25 |
| Graft source (BM vs PBSC) | 1.26 (0.69–2.30) | 0.44 | 1.03 (0.49–2.16) | 0.94 | 1.52 (0.88–2.65) | 0.14 | 0.88 (0.44–1.78) | 0.73 |
| Gender Male vs Female | 1.15 (0.94–1.41) | 0.18 | 1.15 (0.90–1.48) | 0.26 | 1.14 (0.93–1.40) | 0.20 | 1.00 (0.80–1.25) | 1 |
| MAC vs RIC | 0.95 (0.77–1.17) | 0.63 | 1.14 (0.89–1.46) | 0.29 | 0.93 (0.76–1.14) | 0.48 | 1.42 (1.14–1.77) | 0.002 |
| Stage (CR1 vs later) | 0.98 (0.74–1.31) | 0.91 | 1.20 (0.83–1.71) | 0.33 | 0.87 (0.66–1.14) | 0.32 | 0.91 (0.67–1.23) | 0.55 |
| Donor group (matched vs mismatched) | 1.54 (1.19–1.98) | <0.001 | 1.38 (1.01–1.89) | 0.041 | 1.54 (1.20–1.97) | <0.001 | 1.53 (1.14–2.06) | 0.005 |
| Cytogenetic risk (Adverse vs favorable-intermediate) | 1.57 (1.21–2.04) | <0.001 | 1.22 (0.88–1.70) | 0.23 | 1.67 (1.29–2.16) | <0.001 | 1.26 (0.94–1.69) | 0.12 |
| KPS (90–100 vs <90) | 0.66 (0.48–0.92) | 0.015 | 0.70 (0.46–1.06) | 0.09 | 0.67 (0.49–0.93) | 0.016 | 0.75 (0.52–1.06) | 0.10 |
| HCT CI (≥3 vs 0–2) | 1.71 (1.31–2.22) | <0.001 | 1.99 (1.43–2.77) | <0.001 | 1.60 (1.24–2.07) | <0.001 | 1.07 (0.80–1.43) | 0.65 |
| DRI (2–3 vs 0–1) | 2.11 (1.59–2.79) | <0.001 | 1.73 (1.20–2.48) | 0.003 | 2.20 (1.67–2.89) | <0.001 | 1.42 (1.03–1.97) | 0.033 |

**Table 3. Multivariate analysis of statistically significant predictors:** NRM, nonrelapse mortality; RFS, relapse-free survival; GRFS, GvHD-free relapse-free survival; DRI, disease risk index; HCT-CI, hematopoietic cell transplant comorbidity index; CR1, complete remission 1; HR, hazard ratio.

| Predictor | Mortality | | NRM | | RFS | | GRFS | |
|---|---|---|---|---|---|---|---|---|
| | HR (95% CI) | p value | HR (95% CI) | p HR (95% CI) value | HR (95% CI) | p value | HR (95% CI) | p value |
| Opioid User (Yes vs No) | 2.32 (1.50–3.59) | <0.001 | 1.88 (1.08–3.29) | 0.026 | 2.06 (1.34–3.16) | 0.001 | | |
| Group age at BMT (18–39 vs ≥60) | 0.37 (0.23–0.58) | <0.001 | 0.28 (0.15–0.53) | <0.001 | 0.42 (0.27–0.64) | <0.001 | | |
| Donor group (matched vs mismatched) | | | 1.75 (1.15–2.66) | 0.009 | 1.67 (1.18–2.35) | 0.004 | 1.66 (1.16–2.37) | 0.005 |
| Cytogenetic risk (Adverse vs favorable-intermediate) | 1.96 (1.42–2.70) | <0.001 | | | 1.90 (1.35–2.67) | <0.001 | | |
| HCT CI (≥3 vs 0–2) | 1.61 (1.20–2.16) | 0.002 | 2.07 (1.48–2.89) | <0.001 | | | | |
| DRI (2–3 vs 0–1) | | | | | 1.54 (1.09–2.16) | 0.013 | 1.42 (1.03–1.96) | 0.035 |

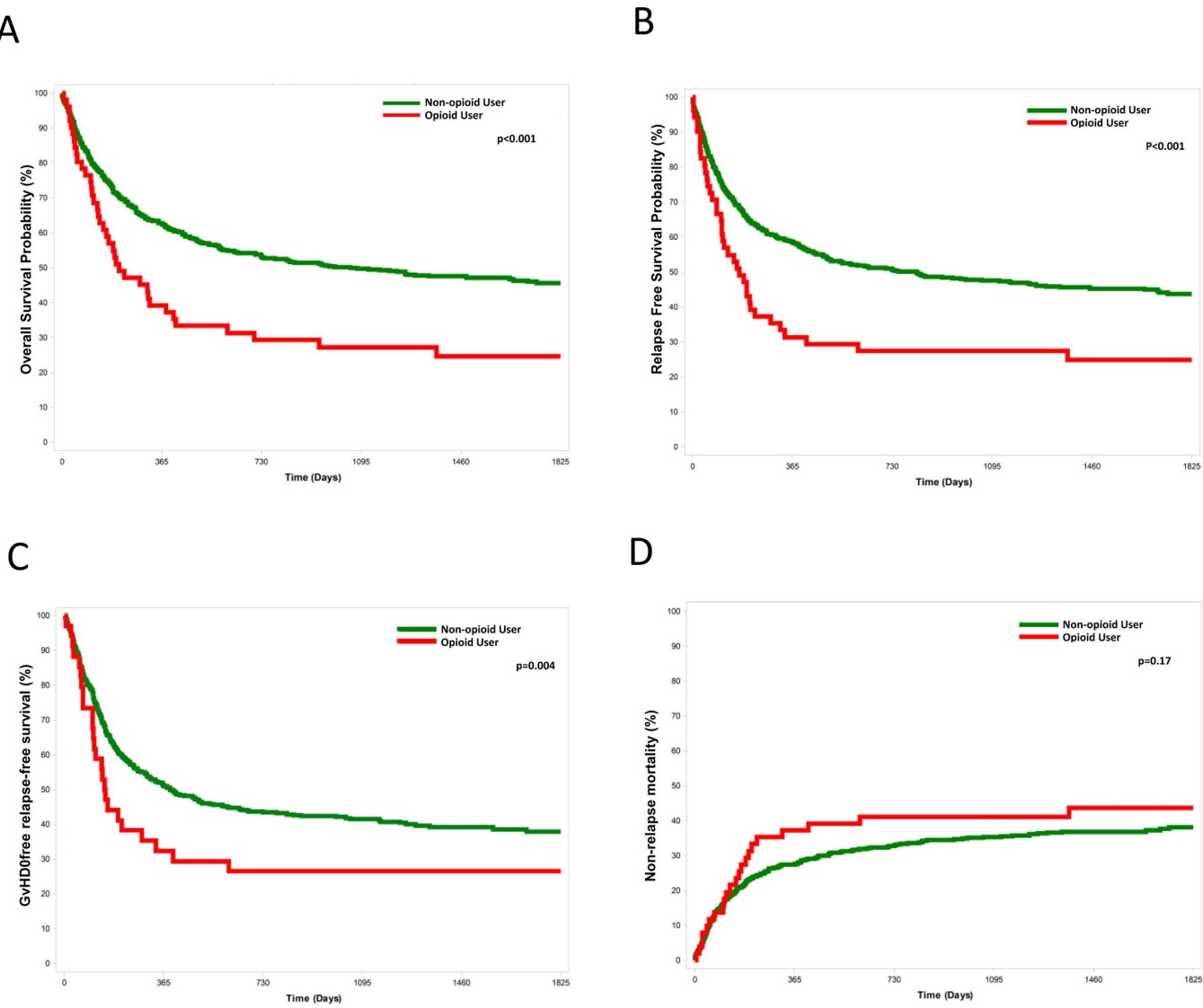

**Fig 2.** A. Overall survival at two years 53% (49.2–57%) for control group vs 29.4% (19.2–45, p < 0.001) for opioid user group; B, relapse free survival at two years was 52.5% (48.7–56.5) for control group and 29.4% (19.2–45%, p < 0.001) for opioid user group; C, GRFS for the opioid user group was 32% (28.6–35.9) for the control cohort and 15.7% (8.3–29.6, 0.004) for the opioid user group D, NRM for IOU 37.3% (25.9–53.5) vs non-IOU 27.5% (24.2–31.2, p = 0.17).

## OS

Fig 2 summarizes the main outcomes. The two-year overall survival (OS) was 53% (95% CI 49.2–57%) for non-IOU group and 29.4% (95% CI 19.2–45%) for IOU group (HR 1.77, 95% CI 1.26–2.48, p < 0.001) (Fig 2A). Median survival was 1,055 days (95% CI 712–1738) for non-IOU patients and 208 days (95% CI 151–415) for IOU patients.

In the multivariate analysis, significant factors included IOU status, age, donor type, cytogenetic risk, and HCT-CI score (Tables 2 and 3). Among the IOU patients, the cause of death was relapse in 22 cases (59.4%), with the remaining 15 (40.1%) attributed to NRM (GVHD, sepsis or organ dysfunction). In the non-IOU group, 106 patients (32.42%) died from infection, 106 patients (32.42%) experienced relapse, and 66 (20.18%) died from GVHD.

### NRM, CIR and RFS

At one-year, non-relapse mortality was 37.3% (95% CI 25.9%–53.5%) for IOU group and 27.5% (95% CI 24.2%–31.2%) for non-IOU group (p = 0.17, Fig 2D). Factors independently associated with NRM comprised IOU status, age, and donor type (Tables 2 and 3). Relapse free survival at two years was 52.5% (48.7–56.5) for the control group and 29.4% (19.2–45%, p < 0.001) for the opioid user group (Fig 2B).

Importantly, after 2 years, 24 patients (31.4%, 95% CI 20.8%–47.4%) of the IOU patients experienced relapse, and 132 (16.4%95% CI 13.7%–19.5%) of the non-IOU relapsed, p = 0.0049. The median time for relapse was 147 days in the IOU group (range 52–393) and 209 day for non-IOU (range 96–1793), p = 0.0082. In the analysis of factors independently associated with LFS, IOU status, age, donor type and cytogenetic risk were found to be significant (Tables 2 and 3).

### aGVHD, cGVHD and GRFS

At 100 days, the cumulative incidence of grade III-IV aGVHD, was 5.9% (95% CI 1.9%–17.9%) in the IOU group versus 4% (95% CI 2.7%–5.9%) in the non-IOU group, p = 0.5281.

At 2 years, the cumulative incidence of moderate to severe cGVHD was 42.2% (95% CI 26.1%–68.3%) in the IOU group versus 33% (95% CI 28.8%–37.9%) in the control group, p = 0.4352.

GRFS for the opioid user group it was 32% (28.6–35.9) for the control cohort and 15.7% (8.3–29.6, p = 0.0043) for the opioid user group (Fig 2C).

In the multivariate analysis, IOU status was not significantly associated with GRFS, but donor type and DRI were significant (Tables 2 and 3).

## Discussion

The present study observed that the intense use of opioids appears to be associated with significantly poorer outcomes following alloHCT, and opioid use for more than 30 consecutive days was linked to worse overall survival after two years, even after adjusting for competing events in the MVA. To our knowledge, this represents the largest cohort of alloHCT patients studied for opioid use and the first to report an correlation between increased relapse incidence and worse survival among IOU.

The prescription of opioid medications has increased significantly in North America in recent years, particularly for non-cancer pain [19]. Between 2015 and 2018, 6% of the US adult population reported opioid use, and in 2020, 143 million prescriptions were dispensed [9]. In contrast, other regions such as Africa, Asia, and Central and South America have lower consumption [19]. According to recent data, in 2020 there were 97,799 overdose related deaths in the United States with an age adjusted rate of 28.3 per 100 000 standard population [5]. Opioid misuse disproportionately affects ethnic minority groups, adolescents, unemployed and uninsured individuals [20–22], and these social determinants of health are linked to worse outcomes following alloHCT [23–25]. Adverse effects of opioids include tolerance, addiction and side effects such as respiratory depression, nausea and constipation [3]. Prolonged opioid use has also been associated with the onset or recurrence of depression [26], and overlapping opioid prescriptions increase the risk of opioid overdose [27].

The long-term use of opioids has been associated with increased 5 year mortality, regardless of the underlying illness [28]. Several studies have demonstrated that the use of opioid therapy for chronic pain after solid organ transplantation is a predictor of adverse events in this population [29–31]. Similarly, high-dose opioid use to manage cancer-related pain, both postoperatively and in patients with solid tumors, is linked to reduced survival [32–34]. In

patients undergoing palliative care for cancer, the use of high-dose opioids is often necessary to control symptoms, and in hospice patients, opioid doses are typically increased until the pain is controlled without hastening end-of-life [35,36]. However, many providers bypass the second step of the WHO analgesic ladder and begin treatment with a strong opioid at a lower dose [37] for patients with hematological malignancies.

The literature on the impact of opioid use in alloHCT remains scant. In the autologous stem cell transplant (ASCT) setting. Farrukh et al. reported the outcomes of 736 patients with hematological malignancies who underwent alloHCT and survived beyond 2 years [38]. Among these long-term survivors, 39.4% reported experiencing pain, with a 2.6-fold increased likelihood of pain compared to controls. Sweiss et al. reported the outcomes of 174 patients who underwent ASCT for multiple myeloma [39]. The study defined "chronic opioid users" as patients with an active opioid prescription for more than three months. Of the entire cohort, 92 patients (52.9%) were identified as chronic opioid users. In multivariate analysis, chronic opioid users had significantly worse overall survival (HR 6.7, 95% CI 1.73–26.09; p = 0.006), although no significant difference was observed in terms of progression-free survival.

The same group reported on chronic opioid use in the alloHCT setting [40], analyzing the outcomes of 159 alloHCT patients. A total of 149 patients (93.7%) received opioids, with chronic opioid use documented in 38 patients (23.9%). Notably, only 23 of these patients (60%) had documented indication for the use of analgesia in their medical record. The group observed that the chronic use of opioids prior to transplant was associated with a worse overall survival in the multivariate analysis (HR 2.99, 95% CI 1.59–5.64; p = 0.001).

Opioids influence cancer cells and anti-tumor immunity through multiple pathways, including toll-like receptors, the sympathetic nervous system, and the hypothalamic-pituitary-adrenal axis [41]. Several in vitro and in vivo studies have shown that opioids can be immunosuppressive [42], which is of relevance when considering the graft-versus-leukemia effect and its pivotal importance as a curative mechanism in diseases such as AML. Morphine and other opioids have been shown to impair the function of macrophages, natural killer (NK) cells, and T-cells, reducing their ability to clear pathogens and regulate immune responses [43]. The activation of the hypothalamic-pituitary-adrenal axis and the sympathetic nervous system by opioids has been implicated in reduced NK cell cytotoxicity and altered cytokine production, further compromising immune defense mechanisms [42].

These pathways can lead to decreased immune cell infiltration into the tumor microenvironment, reduced phagocytosis, and diminished cytotoxic activity of NK cells, potentially exacerbating tumor progression [44]. Importantly, studies have demonstrated that opioid-induced immune modulation is not uniform across all opioids; for instance, buprenorphine has been found to have a lesser impact on immune suppression compared to morphine [42].

While systemic opioid use has been associated with shorter survival in cancer patients and increased susceptibility to infections, definitive causality has yet to be established [45].

Given this context, it is tempting to speculate that the higher relapse incidence observed in the IOU group could partly be due to the potentially immunosuppressive properties of opioids, which may weaken the graft-versus-leukemia effect.

We acknowledge that the observed association between opioid use and outcomes in alloHCT patients is correlational and does not establish causation. It is possible that opioid use followed significant medical events, such as relapse, rather than contributing to them. This possibility highlights the need for caution in interpreting these findings and the importance of further research. While the correlation observed raises questions about whether opioid use could have potential negative consequences in this vulnerable population, it is beyond the scope of this study to establish causality. Prospective studies are needed to elucidate the complex interplay between opioid use and outcomes in allogeneic stem cell transplantation patients.

The retrospective nature of our study is a significant limitation, as is the presence of unmeasured confounders such as social determinants of health (educational level, alcohol and illicit drug use, employment status, among others) that were not collected in our dataset, which could have influenced outcomes in the IOU group. In addition, reliance on pharmacy fill records does not guarantee that the patients were adherent to the opioid prescriptions. ICES privacy policies further restricted our ability to provide detailed outcomes experienced for patients with small sample sizes (6 or fewer), limiting some specifics in the results section.

Another limitation of our methodology involves the calculation of relapse-free survival (RFS). Patients who initiated opioid use more than one-year post-transplant were excluded to minimize survival bias. This decision was based on the rationale that patients surviving beyond one year without requiring opioids may represent a distinct subgroup with inherently better outcomes. Including these patients could have introduced confounding factors, potentially obscuring the relationship between early opioid use and relapse or survival outcomes. While this approach allowed for a more homogeneous analysis during the critical first year post-transplant—a period marked by heightened risks of immune reconstitution-related complications, including relapse—it limits the generalizability of our findings to late-onset opioid users.

In conclusion, our study suggests a negative association between intense opioid use and overall survival following allogeneic stem cell transplantation. Given the widespread use of opioids in the alloHCT setting, safer pain management regimens should be sought in order to reduce toxicity while ensuring effective pain control.

## Supporting information

**S1 File. Supplementary information.**
(DOCX)

## Author contributions

**Conceptualization:** Tommy Alfaro Moya, Abel Santos Carreira, Jonas Mattsson.

**Data curation:** Tommy Alfaro Moya.

**Formal analysis:** Tommy Alfaro Moya, Shiyi Chen, Mats Remberger.

**Supervision:** Mats Remberger, Refik Saskin.

**Writing – original draft:** Tommy Alfaro Moya, Abel Santos Carreira, Jonas Mattsson.

**Writing – review & editing:** Tommy Alfaro Moya, Refik Saskin, Igor Novitzky-Basso, Jonas Mattsson.

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
