## [Decision Letter · Decision Letter 0]

20 Nov 2024

PONE-D-24-36417Potential Impact of Opioid Use in the Outcomes of Allogeneic Hematopoietic Stem Cell Transplantation: Increased Incidence of Relapse and Worse Survival Among Intense Opioid UsersPLOS ONE

Dear Dr. Mattsson,

Thank you for submitting your manuscript to PLOS ONE. After careful consideration, we feel that it has merit but does not fully meet PLOS ONE’s publication criteria as it currently stands. Therefore, we invite you to submit a revised version of the manuscript that addresses the points raised during the review process.

We look forward to receiving your revised manuscript.

Kind regards,

Monia Marchetti

Academic Editor

PLOS ONE

Journal Requirements:

“This study contracted ICES Data & Analytic Services (DAS) and used de-identified data from the ICES Data Repository, which is managed by ICES with support from its funders and partners: Canada’s Strategy for PatientOriented Research (SPOR), theOntario SPOR Support Unit, the Canadian Institutes of Health Research and the Government of Ontario. This study was supported through provision of data by ICES and Cancer Care Ontario (CCO) and through funding support to ICES from an annual grant by the Ministry of Health (MOH) and the Ontario Institute for Cancer Research (OICR). The opinions, results and conclusions reported in this paper are those of the authors. No endorsement by ICES or any of its funders or partners is intended or should be inferred.”

4. In this instance it seems there may be acceptable restrictions in place that prevent the public sharing of your minimal data. However, in line with our goal of ensuring long-term data availability to all interested researchers, PLOS’ Data Policy states that authors cannot be the sole named individuals responsible for ensuring data access (http://journals.plos.org/plosone/s/data-availability#loc-acceptable-data-sharing-methods).

5. Please ensure that you refer to Figure 2 in your text as, if accepted, production will need this reference to link the reader to the figure.

Reviewers' comments:

Reviewer's Responses to Questions

**Comments to the Author**

1. Is the manuscript technically sound, and do the data support the conclusions?

Reviewer #1: No

Reviewer #2: Partly

2. Has the statistical analysis been performed appropriately and rigorously? 

Reviewer #1: No

Reviewer #2: Yes

3. Have the authors made all data underlying the findings in their manuscript fully available?

Reviewer #1: Yes

Reviewer #2: Yes

4. Is the manuscript presented in an intelligible fashion and written in standard English?

Reviewer #1: No

Reviewer #2: Yes

5. Review Comments to the Author

Reviewer #1: - The manuscript was submitted with tracked changes and comments, making it very difficult and unpleasant to read. This feels more like a draft version than a completed manuscript.

- All predictors in Cox, Kaplan-Meier and Grey's analysis must either be known at time 0 (transplant in this case), or be treated as time dependent covariates (which is only available for Cox).

- The authors observe a correlation and not causation, but make claims toward the latter. The use of opioids may have actually followed a significant medical problem (for example, a relapse) and not the other way around. Therefore, the conclusions are unfounded.

Reviewer #2: I congratulate the authors for this rare mentioned title on opiate use in alloHCT. I appreciate the methodology and idea behind it, but strong conclusive remarks and correlating opiate use with relapse and dismal outcome needs more clarification. My comments are shared below;

Major:

1- Title: Too strong, did we have any clue or proıf how to explain this phenomenon beyond immunosupressive effect of opiates.

2- Aim: Since 30 years I am a clinical transplanter I hardly remember any patient of mine, except autologous myeloma patients , and patients with avascular necrosis or severe osteoporosis needs opiate prescriptions. I am not convinvrd why these patients are on opiates.

3- Study population: Do patients started opiate use or the physcisians , it is not clear in the paper. The paper exludes before hand patients started opiates after 1 year post allo, and underlines the bias for relapse vulnerable period. How do they know that opiates will have impact on relapse beforehand? This is a hypothesis and should be performed in all opiate users, and selective analyses can be performed on selected populations.

4- Definition: According which reference or infobase you established the definition of IOU. Why you did not include the non malignant diseases? They suffer for certain degree transplant complications. The exclusion of benign disorders means that your intent is not only alloHCT base but malignant hematology based.

5- Methodology for relapse: For RFS we use 2 years threshold but you did not consider patients starting to use opiates after 1 year, this approach is erratic.

6- How do you biologically prove without knowing any exact dose, rason for prescription etc. that intense use is correlated with poor outcome. This is an observation of a retrospective single center cohort which has to be taken very cautiously.

7- L409 "Given the widespread use of opioids in the alloHCT setting", is this correct, do we have any clear data or publication so far since 60 years of alloHCT history.

Minor:

1- T1 please define Molecular and cytogenetics more clearly as caption,

2- T2 please revise as 2a UVA and 2b MVA, in current form not legible, and difficult to evaluate.

3- L380 "Several in vitro studies have shown that opioids can be immunosuppressive", very speculative either share details on pathophysiology or revise.

6. PLOS authors have the option to publish the peer review history of their article (what does this mean? ). If published, this will include your full peer review and any attached files.

**Do you want your identity to be public for this peer review?** For information about this choice, including consent withdrawal, please see our Privacy Policy .

Reviewer #1: No

Reviewer #2: **Yes: ** mutlu arat

---

## [Author Response · Author response to Decision Letter 1]

29 Jan 2025

Response to Reviewers: Manuscript No. PONE-D-24-36417

Journal Requirements:

Comment 1: Please ensure that your manuscript meets PLOS ONE's style requirements, including those for file naming.

Reply: We have reviewed the style requirements and ensured every file is named according to your guidelines.

Response: Headings and subheadings have been modified to meet PLOS ONE’s style requirements.

Comment 2: Thank you for stating the following in the Acknowledgments Section of your manuscript:

“This study contracted ICES Data & Analytic Services (DAS) and used de-identified data from the ICES Data Repository, which is managed by ICES with support from its funders and partners: Canada’s Strategy for Patient-Oriented Research (SPOR), the Ontario SPOR Support Unit, the Canadian Institutes of Health Research, and the Government of Ontario. This study was supported through provision of data by ICES and Cancer Care Ontario (CCO) and through funding support to ICES from an annual grant by the Ministry of Health (MOH) and the Ontario Institute for Cancer Research (OICR). The opinions, results, and conclusions reported in this paper are those of the authors. No endorsement by ICES or any of its funders or partners is intended or should be inferred.”

Reply: We greatly appreciate your thorough review and the guidance provided regarding the placement of funding acknowledgments within the manuscript.

Response: In response, we have made the following changes:

• Funding Statement: Updated to: “This study was supported through the provision of data by ICES and Cancer Care Ontario (CCO), with funding support to ICES from an annual grant by the Ministry of Health (MOH) and the Ontario Institute for Cancer Research (OICR). The authors did not receive any specific funding for this work.” Lines 438-441.

• Acknowledgments Section: Revised to: “This study contracted ICES Data & Analytic Services (DAS) and used de-identified data from the ICES Data Repository, which is managed by ICES with support from its funders and partners: Canada’s Strategy for Patient-Oriented Research (SPOR), the Ontario SPOR Support Unit, the Canadian Institutes of Health Research, and the Government of Ontario. The opinions, results, and conclusions reported in this paper are those of the authors. No endorsement by ICES or any of its funders or partners is intended or should be inferred.” Lines 422-4431

Comment 3: We note that you have indicated that there are restrictions to data sharing for this study. For studies involving human research participant data or other sensitive data, we encourage authors to share de-identified or anonymized data. However, when data cannot be publicly shared for ethical reasons, we allow authors to make their datasets available upon request.

(a) If there are ethical or legal restrictions on sharing a de-identified dataset, please explain them in detail (e.g., data contain potentially identifying or sensitive patient information, data are owned by a third-party organization, etc.) and who has imposed them (e.g., a Research Ethics Committee or Institutional Review Board, etc.). Please also provide contact information for a data access committee, ethics committee, or other institutional body to which data requests may be sent.

(b) If there are no restrictions, please upload the minimal anonymized dataset necessary to replicate your study findings to a stable, public repository and provide us with the relevant URLs, DOIs, or accession numbers.

Reply: Thank you for your inquiry regarding the data-sharing policy from ICES.

(a) Ethical and Legal Restrictions on Data Sharing: The dataset used in this study is held securely in coded form at the Institute for Clinical Evaluative Sciences (ICES). Data-sharing agreements with ICES prohibit the dataset from being made publicly available due to ethical and legal restrictions. These restrictions ensure the privacy and confidentiality of the data, which include sensitive patient information. ICES operates under Ontario’s privacy legislation, which governs the collection, use, and disclosure of health data.

Access to the data can be granted to researchers who meet specific criteria for confidential access as outlined by ICES. Data access requests may be directed to ICES Data & Analytic Services (DAS) via their website: www.ices.on.ca/DAS.

For additional information or data access requests, please contact: ICES Data & Analytic Services Email: das@ices.on.ca

(b) Data Availability Statement Update: Updated to: “The dataset from this study is held securely in coded form at the Institute for Clinical Evaluative Sciences (ICES). While data-sharing agreements prohibit ICES from making the dataset publicly available, access may be granted to those who meet pre-specified criteria for confidential access, available at www.ices.on.ca/DAS. The full dataset creation plan and underlying analytic code are available from the authors upon request, understanding that the computer programs may rely upon coding templates or macros that are unique to ICES and may require modification.” Lines 27-34.

Comment 4: Please ensure that you refer to Figure 2 in your text as, if accepted, production will need this reference to link the reader to the figure.

Reply: Thank you for your feedback.

Response: We have now referred to Figure 2 in the main text of the manuscript to ensure proper linkage and clarity for readers. Line 278.

Review Comments to the Author

Reviewer #1:

Comment: The manuscript was submitted with tracked changes and comments, making it very difficult and unpleasant to read. This feels more like a draft version than a completed manuscript.

Reply: Thank you for bringing this to our attention. We sincerely apologize for the oversight in submitting a version of the manuscript with tracked changes and comments.

Response: A clean, fully revised version of the manuscript, free of tracked changes and comments, has been uploaded for your review.

Comment: All predictors in Cox, Kaplan-Meier, and Grey's analysis must either be known at time 0 (transplant in this case), or be treated as time-dependent covariates (which is only available for Cox).

Reply: Thank you for raising this important point regarding the timing of predictors in our analyses. We would like to clarify our approach and explain how we addressed potential survival bias.

Response: The start time (time 0) for our analysis is not the time of transplant for all patients. As shown in Figure 2, we defined the start time as follows:

• For opioid users, the start time is defined as:

o The first time opioids were used within one year post-BMT if the total duration exceeded 30 days.

o The time of transplant if opioid use was initiated before BMT or within one year post-BMT for less than 30 days, but the patient died within one year while on opioids.

• The median start time for opioid users is 95 days post-transplant.

For the comparator group, the start time was adjusted to account for survival bias due to the delayed initiation of opioids in many patients in the opioid user group. Specifically:

• For patients who did not use opioids, the start time was set to time of BMT + 95 days.

• For patients who started opioids but did not meet the criteria for the opioid user group, the start time was set to time of opioid initiation + 95 days.

All predictors included in the Cox proportional hazards model, Kaplan-Meier analysis, and Grey's test were known at the defined start time (time 0). By aligning the start time for the comparator group with the median start time of the opioid user group (95 days), we accounted for survival bias, ensuring that both groups are analyzed from equivalent time points relative to their risk exposure. We hope this explanation addresses your concern. Please let us know if further clarification is needed. Lines 118-166.

Comment: The authors observe a correlation and not causation, but make claims toward the latter. The use of opioids may have actually followed a significant medical problem (for example, a relapse) and not the other way around. Therefore, the conclusions are unfounded.

Reply: Thank you for this insightful observation. We acknowledge that the use of opioids in our cohort could indeed have followed a significant medical event, such as a relapse, rather than being a contributing factor to it.

Response: That said, our study found that intense opioid use (IOU) was significantly associated with poorer outcomes, including a higher relapse rate at two years (31.4% vs. 16.4%, p=0.0049) and reduced relapse-free survival (29.4% vs. 52.5%, p<0.001). Furthermore, the median time to relapse was significantly shorter in the IOU group (147 days; range 52-393) compared to the non-IOU group (209 days; range 96-1793), with a p-value of 0.0082. Additionally, multivariate analysis indicated that IOU status was independently associated with a 2.32 times higher instantaneous rate of death compared to non-IOU (HR 2.32, 95% CI 1.5-3.5, p=0.002). These findings suggest that IOU status may serve as an important marker for identifying high-risk patients who could benefit from targeted interventions. Lines 298-301.

While we fully agree that correlation does not imply causation, we believe our findings provide an important basis for further exploration. The correlation observed raises critical questions about whether opioid use could have potential negative consequences in this vulnerable population, particularly given existing evidence of the immunosuppressive effects of opioids. It is not our intention to imply causation but rather to suggest that these results warrant further investigation through prospective studies to better understand the complex interplay between opioid use and outcomes in allogeneic stem cell transplantation patients. We have revised the manuscript to ensure that our language reflects the observational nature of the study and avoids any implication of causation. Lines 316-420.

Reviewer #2:

Major Comments

Comment 1: Title: Too strong, did we have any clue or proof how to explain this phenomenon beyond immunosuppressive effect of opiates.

Reply: Thank you for your observation regarding the title. As our analysis focuses on identifying correlations rather than proving causation, we understand the importance of ensuring that the title accurately reflects the study's observational nature.

Response: We have revised the title to: “Exploring the Impact of Opioid Use on Outcomes in Allogeneic Hematopoietic Stem Cell Transplantation.” This updated title better represents the exploratory and observational nature of the work while leaving room for future research to further investigate the mechanisms underlying the observed associations. Lines 1-2.

Comment 2: Aim: Since 30 years I am a clinical transplanter I hardly remember any patient of mine, except autologous myeloma patients, and patients with avascular necrosis or severe osteoporosis needs opiate prescriptions. I am not convinced why these patients are on opiates.

Reply: In North America, opioid prescriptions have historically been prevalent, particularly for managing moderate to severe pain in medically complex populations. For hematopoietic stem cell transplant (HSCT) patients, pain can stem from diverse sources, including mucositis, granulocyte-colony stimulating factor (G-CSF)-associated musculoskeletal pain, therapy-related neurotoxicity, and graft-versus-host disease (GVHD), among other complications. This clinical necessity is compounded by historical and systemic factors contributing to opioid prescribing trends, as highlighted by Alpert et al. (2022), including aggressive marketing and shifts in prescribing practices that significantly increased opioid use across patient populations, including those undergoing cancer and transplant treatments. Lines 93-97.

Response: As noted by Oh et al. (2020), effective pain management is essential for maintaining quality of life during the intensive treatment course of HSCT. Opioids, both short-acting and long-acting, are frequently utilized for addressing severe pain that is unresponsive to other interventions. The National Comprehensive Cancer Network (NCCN) guidelines underscore the importance of appropriate and safe opioid use in cancer-related pain management, particularly in cases of refractory or persistent pain. Moreover, Niscola et al. (2006) describe the dual nociceptive and neuropathic components of pain experienced by HSCT patients, further necessitating the use of opioids in certain cases.

In our cohort, the observed patterns of opioid use likely reflect these broader prescribing practices and the pressing clinical need to manage the multifaceted and often severe pain syndromes associated with HSCT. The disparity in prescribing practices between North America and other regions, as noted by Reviewer 2, was a key motivation for initiating this study. We aimed to investigate and contextualize these differences, which remain a significant concern given the implications for patient care and the ongoing opioid crisis.

Comment 3: Study population: Do patients start opiate use or the physicians? It is not clear in the paper. The paper excludes beforehand patients starting opiates after 1-year post-allo and underlines the bias for relapse vulnerable period. How do they know that opiates will have an impact on relapse beforehand? This is a hypothesis and should be performed in all opiate users, and selective analyses can be performed on selected populations.

Reply: Thank you for these clarifications.

Response:

• Initiation of Opioid Use: In our study, opioid use was initiated based on provider prescriptions in response to clinical indications such as moderate to severe pain related to post-transplant complications, including mucositis, musculoskeletal pain, or GVHD. This reflects standard clinical practice as outlined in pain management guidelines for hematologic malignancies and post-transplant care. We have revised the manuscript to make this clearer. Lines 234-236.

• Exclusion of Patients Initiating Opioid Use Beyond One Year Post-Transplant: The rationale for excluding patients who began opioid use more than one-year post-transplant was based on our aim to study the impact of opioid use during the critical period of immune reconstitution and relapse vulnerability. Studies indicate that this period is characterized by significant risks for complications that could necessitate opioid use. However, we acknowledge that this criterion may introduce selection bias, as it limits the generalizability of our findings to early post-transplant opioid users. Lines 406-415.

• Hypothesis Testing and Selective Analyses: You are correct that the study is hypothesis-driven, exploring whether opioid use during this vulnerable period might correlate with an increased incidence of relapse. While our focus was on early opioid use, we agree that extending the analysis to all opioid users—regardless of the time of initiation—would provide a more comprehensive understanding of the relationship between opioid use and relapse. Selective analyses could then be performed on subpopulations, such as late-onset opioid users, to explore potential differences in outcomes. We will emphasize this limitation in the revised manuscript and highlight the need for future studies to include a broader range of opioid users. Lines 406-415.

Comment 4: Definition: According to which reference or infobase you established the definition of IOU. Why did you not include the non-malignant diseases? They suffer for a certain degree of transplant complications. The exclusion of benign disorders means that your intent is not only alloHCT-based but malignant hematology-based.

Reply: Thank you for your thoughtful comment regarding the definition of intense opioid use (IOU) and the exclusion of patients with non-malignant diseases. We appreciate the opportunity to clarify our rationale and approach.

Response:

• Definition of Intense Opioid Use (IOU): The definition of

---

## [Decision Letter · Decision Letter 1]

2 Mar 2025

Exploring the impact of opioid use on outcomes in allogeneic hematopoietic stem cell transplantation

PONE-D-24-36417R1

Dear Dr. Mattsson,

We’re pleased to inform you that your manuscript has been judged scientifically suitable for publication and will be formally accepted for publication once it meets all outstanding technical requirements.

Kind regards,

Monia Marchetti

Academic Editor

PLOS ONE

Additional Editor Comments (optional):

the authors fulfilled the reviewers request and the manuscript can be published in its present format

Reviewers' comments:

Reviewer's Responses to Questions

**Comments to the Author**

1. If the authors have adequately addressed your comments raised in a previous round of review and you feel that this manuscript is now acceptable for publication, you may indicate that here to bypass the “Comments to the Author” section, enter your conflict of interest statement in the “Confidential to Editor” section, and submit your "Accept" recommendation.

Reviewer #1: All comments have been addressed

Reviewer #2: All comments have been addressed

2. Is the manuscript technically sound, and do the data support the conclusions?

Reviewer #1: Yes

Reviewer #2: (No Response)

3. Has the statistical analysis been performed appropriately and rigorously? 

Reviewer #1: Yes

Reviewer #2: (No Response)

4. Have the authors made all data underlying the findings in their manuscript fully available?

Reviewer #1: Yes

Reviewer #2: (No Response)

5. Is the manuscript presented in an intelligible fashion and written in standard English?

Reviewer #1: Yes

Reviewer #2: (No Response)

6. Review Comments to the Author

Reviewer #1: (No Response)

Reviewer #2: (No Response)

7. PLOS authors have the option to publish the peer review history of their article (what does this mean? ). If published, this will include your full peer review and any attached files.

**Do you want your identity to be public for this peer review?** For information about this choice, including consent withdrawal, please see our Privacy Policy .

Reviewer #1: No

Reviewer #2: No

---

## [Editor Report · Acceptance letter]

PONE-D-24-36417R1

PLOS ONE

Dear Dr. Mattsson,

I'm pleased to inform you that your manuscript has been deemed suitable for publication in PLOS ONE. Congratulations! Your manuscript is now being handed over to our production team.

Kind regards,

on behalf of

Dr. Monia Marchetti

Academic Editor

PLOS ONE